# OpenReview forum: "Evaluating the Moral Beliefs Encoded in LLMs"
_NeurIPS.cc/2023/Conference — NeurIPS 2023 spotlight_

### Official Review · Reviewer_tn2x · 2023-07-05

**Soundness:** 3 good
**Presentation:** 2 fair
**Contribution:** 2 fair
**Rating:** 6
**Confidence:** 4

**Summary:**

The paper studies the encoded moral norms of various language models. To this end, the authors first describe different measurements: semantic likelihood, consistency, and uncertainty. To construct the dataset to analyze LMs, the authors chose to use OpenAI’s text-davinci-003 and GPT-4 to create the content of the survey. This content is reviewed by the authors of the paper leading to the MoralChoice Dataset containing high-ambiguity and low-ambiguity samples. The scenarios are annotated using Surge AI. Using the introduced dataset and the described metrics, the present paper analyzes various LLMs. However, instead of benchmarking a model’s performance, the present study analysis differences between LMs.

**Strengths:**

- The paper addresses the important topic on how moral norms are encoded in LLMs.
- The paper introduce a novel dataset
- Extensive comparison of various open-source and commercial LLMs.
- Detailed description of applied metrics.

**Weaknesses:**

- Missing details on how the introduced dataset is annotated. What is Surge AI? How is the number of annotators justified? What is the demographic background of the annotators?

- Many statements, such as "These scenarios mimic the real-world situation where users turn to LLMs for advice.“ require an ethical discussion in a dedicated section.

- The authors selected a temperature of 1 to sample from the LLMs, which decreases the confidence ("riskier“ answers) and greater variability of the model and leads to non-deterministic behavior. The authors, unfortunately, do not justify these decisions. Since the present study measures the uncertainty of models’ outputs, the sampling process is expected to have a high impact on the results. Furthermore, many of the investigated systems do not allow configuring the sampling process, which results, at least on Claude and GPT-4, in the limited significance of the presented observations and comparison.

- Unfortunately, the discussion of the results is very limited. A more extensive discussion apart from the second paragraph in the discussion section would help to increase the impact of the present study.

*Minor Comments:*
 - Typo in line 200: To help …
- Inconsistent capitalisation of surge AI
- Missing References
    - Schramowski et al. 2022 (https://www.nature.com/articles/s42256-022-00458-8)
    - Ganguli et al. 2023 (https://arxiv.org/pdf/2302.07459.pdf)
    - Hämmerl et al. 2023 (https://arxiv.org/abs/2211.07733)

**Questions:**

- Can you provide more details about the annotation process (see above)?
- Can you further elaborate on your statement: "This suggests the need for future research to focus on diversifying human alignment data or providing explicit moral guidance during training.“ Does this apply to the current alignment tuning of the commercial systems anthropic/openai models or only on open source models?
- The commonsense subset of Hendrycks [30] ethics dataset seems very similar to the introduced dataset. Why did you choose to construct a new dataset?

**Limitations:**

The applied sampling should be justified and listed as limitations. Studying different sampling configurations, such as in [1], would further strengthen the paper.

---

> ### Author Rebuttal · Authors · 2023-08-10
>
> We thank the reviewer for their thoughtful feedback and questions. We have actioned minor comments on typos and references and address the main concerns below.
>
> **(1) Missing details on how the introduced dataset is annotated. What is Surge AI? How is the number of annotators justified? What is the demographic background of the annotators?**
>
> Please see `Lack of Annotator Details` in the general response. It is included in Appendix A3.
> Surge AI is an annotation company https://www.surgehq.ai/.
> We chose three because the annotator agreements were high. Appendix A.5 reports the annoator agreement.
>
> **(2) Many statements, such as "These scenarios mimic the real-world situation where users turn to LLMs for advice.“ require an ethical discussion in a dedicated section.**
>
> We thank and agree with the reviewer that statements like these are unnecessarily controversial. We do not intend to make statements about whether people use LLMs for ethics advice. Throughout the paper, we have deleted statement like this and focus our exposition on introducing the dataset, methods, and the survey findings. Additionally, we added a paragraph discussing bias in the survey (See `Potential Cultural Bias in the survey`) and a discussion on limitation of prompt-based probes (See `Limitations of Prompt-based Probe`)
>
>
>
> **(3) Authors selected a temperature of 1 to sample from the LLMs, which decreases the confidence ("riskier“ answers) and greater variability of the model and leads to non-deterministic behavior. The authors, unfortunately, do not justify these decisions. Since the present study measures the uncertainty of models’ outputs, the sampling process is expected to have a high impact on the results.**
>
> We thank the reviewer for the discerning question! Choosing such a decoding scheme allows us to access the “original” probabilities encoded in the LLM and to compute the metrics specified in section 2. Choosing different parameters would distort the underlying probability distribution and bias the generated responses. We have additionally included an ablation study in the rebuttal document. Please see `Decoding Method` in the general response.
>
> **(4) Many of the investigated systems do not allow configuring the sampling process, which results, at least on Claude and GPT-4, in the limited significance of the presented observations and comparison.**
>
> This is a great point!  We have highlighted this issue in the experiment section.
>
> **(5) Discussion of the results is very limited. A more extensive discussion apart from the second paragraph in the discussion section would help to increase the impact of the study.**
>
> We thank and agree that more discussion of the results can greatly improve the paper. In addition to the sections in the general response. We included the following paragraphs:
>
> The findings in high-ambiguity scenarios reveal that certain LLMs reflect distinct preferences, even in situations where there is no clear answer. We identify a cluster of models that have high agreement, these correspond to most of the shelf models from OpenAI and Anthropic.
> We hypothesize that it is because these models have been through an ``alignment with human preference" process at the fine-tuning stage. We suspect that these models have been trained on human preference data, where they learned a reward function that then extrapolated it.
>
> Finding in across dataset suggests that variations in the prompt format can greatly influence the response distribution. Specifically, these correspond to the early OpenAI models. These models have been supervised fine-tuning rather than RLHF. We suspect this is because the model pick up spurious correlations in the finetuning dataset.
>
> However, we emphasize this is merely our hypotheses. These models are not public. It’s unclear what data they have been trained on. Many factors could lead to this observation.
> Verifying these hypothesis would require detailed account these models that capture all the confounding factors or controlled experiment and comparison between fixed pre-trained models and RLHF models. Understanding the factors that drive this consensus among the models is a crucial area for future research.
>
> **Can you further elaborate on your statement: "This suggests the need for future research to focus on diversifying human alignment data or providing explicit moral guidance during training.“ Does this apply to the current alignment tuning of the commercial systems anthropic/openai models or only on open source models?**
>
> Great question! This is comment regarding to early OpenAI models (ada/babbage/curie/davinci). We found that these models have low semantic entropy in responses given a fixed prompt templates, but different distribution of responses across templates. We suspect this is because the model pick up spurious correlations in the finetuning dataset. Where the model learned to answer that’s consistent with the annotators when given a prompt format, but does not generalize. We agree that this should be further explained and have revised the draft to reflect that.
>
> **Commonsense subset of Hendrycks ethics dataset seems very similar to the introduced dataset. Why did you choose to construct a new dataset?**
>
> That's a great question! Initially, we considered using the ethics dataset. However, we observed that the examples are relatively short, and certain examples appear arbitrary. For instance, in the Common Sense dataset, there is an example labeled "I crushed the frog" with the label "Not common sense." Furthermore, we have concerns about the models that have already been trained on this dataset. We cannot be certain these models have not been directly trained on this dataset. The concerns led us to design our own datasets.
>
> **Applied sampling should be justified and listed as limitations. Studying different sampling configurations, such as in [1], would further strengthen the paper.**
>
> Thank you for the suggestions! Please see `Decoding Method` in the response.

---

> > ### Comment · Reviewer_tn2x · 2023-08-15
> > **Thank you for the response.**
> >
> > Thank you for the extensive rebuttal. Especially the additional experiments with different sampling configurations improved the paper. Furthermore, the additional discussions on limitations, which should be included in the revised paper, address my concerns. Therefore I'm happy to raise my score.

---

### Official Review · Reviewer_JwcR · 2023-07-14

**Soundness:** 3 good
**Presentation:** 4 excellent
**Contribution:** 3 good
**Rating:** 7
**Confidence:** 5

**Summary:**

The paper studied the moral beliefs encoded in large language models. For this purpose, the authors designed a systematic survey-based framework consisting of three novel metrics, i.e., semantic likelihood, consistency and uncertainty, and the MoralChoice dataset with 680 ambiguous moral scenarios and 685 less ambiguous ones. Based on this framework, they evaluated 24 mainstream LLMs (both open and closed-sourced) to elicit their moral beliefs, focusing on common sense reasoning and agreement degree. The results indicate that while LLMs can perform common-sense moral reasoning, their consistency and certainty vary, highlighting the need for diversifying human alignment data or providing explicit moral guidance during training.

**Strengths:**

* This paper focuses on a crucial topic: moral beliefs in LLMs. While there are already some existing papers in this area (Simmons, 2022; Jin et al., 2022; Arora et al., 2023), to my knowledge, this paper is the first to utilize a relatively large-scale dataset based on an established moral theory. Furthermore, the entire paper is clearly written and well-organized.
* The whole paper is well motivated by the two interesting questions posed in Sec.1, i.e., LLMs' commonsense reasoning and their responses to scenarios with no clear answers. The proposed framework, dataset and metrics are convincing and could well respond to the two questions. Using consistency and uncertainty to unpack the underlying moral beliefs is helpful.
* The experiments and analysis are exhaustive and relatively convincing. The obtained conclusions in Sec.4 could benefit the research of LLM ethics and the development of future LLMs.

#### Refs:
* Simmons, Moral mimicry: Large language models produce moral rationalizations tailored to political identity. 2022.
* Jin et al., MoCa: Cognitive Scaffolding for Language Models in Causal and Moral Judgment Tasks. 2022.
* Arora, et al., Probing Pre-Trained Language Models for Cross-Cultural Differences in Values. In Proceedings of C3NLP. 2023.

**Weaknesses:**

* My biggest concern is the truthfulness of the evaluation results. How can we guarantee that these survey questions really reveal the intrinsic beliefs of LLMs? Models can lie (so do humans), and one recent paper (Perez et al., 2022) demonstrated that LLMs might exhibit sycophancy. I understand that the consistency and uncertainty metrics seem a kind of solution for this problem, but two further limitations exist:
    * Only three kinds of prompt variants, $z$, (line 242) are insufficient. A low coverage of $z$ could lead to a high variance of the metrics involving different $z$. For example, for Eq.(4), $H(c|z)=-\mathbb{E}_{p(z)}[p(c|z)\log p(c|z)]$. Only three variants of $z$ might cause an unreliable estimation of this expectation. The author should report the variance of results among the three prompt variants or consider more variants. For example, use automatic methods to obtain them, like syntax-controlled paraphrase generation (Yang et al., 2022).
    * The authors should report results on different decoding settings. All the results are drawn based on a fixed setting. Since different decoding methods (e.g., top-p, top-k or both) and hyper-parameters (e.g., p, k, temperature and number of samples) can influence the results in some relevant tasks, e.g., fairness evaluation (Dhamala et al., 2022), could the conclusions still hold with different decoding settings?

* Another concern is the reliability of the dataset and annotation.
    * The unclear annotator demographics. Since the whole dataset, especially the favourable actions, are automatically generated (Appendix A) and manually checked (line 193), and the authors used these actions as 'commonsense', the diversity and coverage of annotators should be guaranteed. If the favourable actions are agreed upon by limited annotators (e.g., three white Americans), claiming them as 'commonsense' is problematic.
    * The quality of the initial 100 scenarios needs to be verified. As mentioned in line 628, these scenarios are written by the authors themselves. It's unknown whether the authors have enough knowledge of ethics to write correct and high-quality scenarios.

Generally, this is a good paper. If the authors can address my concerns above, I will consider changing my scores.

#### Refs:
* Perez et al., Discovering language model behaviours with model-written evaluations. 2022.
* Yang et al., Learning Structural Information for Syntax-Controlled Paraphrase Generation. 2022.
* Dhamala et al., An analysis of the effects of decoding algorithms on fairness in open-ended language generation. In 2022 IEEE Spoken Language Technology Workshop (SLT).

**Questions:**

* Does the common morality framework contain only ten rules, or do you just use a part of them?
* Would you release your dataset?
* Have you analyzed the impact of alignment methods on the conclusion, e.g., RLHF and Supervised Fine Tuning besides model scale?
* Why didn't you use GPT-4 to expand the dataset (line 183)?
* Have you tried to evaluate the semantic consistency using perturbations on scenarios (e.g., different expressions) besides the instructions (prompts)?

**Limitations:**

The authors have discussed the limitations in Sec.5 and Appendix C.2.1. However, the authors should also include more discussions of limitations, like limited coverage and ethical considerations.

---

> ### Author Rebuttal · Authors · 2023-08-10
>
> We thank the reviewer for the thoughtful questions and constructive feedback. We have incorporated the references into the revised draft and will address the concerns point by point
>
> **(1) How can we guarantee that these survey questions really reveal the intrinsic beliefs of LLMs?**
>
> Please see `Model Faithfulness` and `Limitations of prompt-based probe` in the general response.
>
> **(2) Only three kinds of prompt variants are insufficient. …**
>
> Thank you for raising this insightful point. The semantic consistency metric was specifically devised to measure the influence of different prompt formats over output variation. In appendix C1, we illustrated the variance of the results under each prompt format. It show most of the variation comes from the compare template.
>
> We agree that three prompts templates are not sufficient to validate whether a model would “select” an action given a scenario. In an ideal scenario, we would indeed explore a broad range of prompt templates or even consider automated paraphrasing approaches. However, practical constraints prevail. Generating and verifying paraphrases entail substantial costs, exacerbated by LLM hallucination susceptibility, demanding meticulous manual verification.
>
> Expanding datasets introduces challenges as well. Our current framework involves three prompt templates, two question orderings per template, and 5-10 samples for each question. Expanding prompt templates will lead to rapid growth of inference time and cost.
>
> The issue of defining the distribution of p(z) also surfaces. An automated approach introduces ambiguity in p(z) distribution, complicating the interpretation of inconsistent model behavior. The controlled nature of prompt templates offers clarity on the source of variations. As demonstrated in Appendix C1, our analysis pinpointed variations to a single prompt template.
>
> In our revised draft, we have clarified the challenges tied to defining p(z) and emphasized the limitations..
>
> **(3) The authors should report results on different decoding settings.**
>
> Please see `Decoding Method` in the general response.
>
> **(4) Unclear annotator demographics. … If the favourable actions are agreed upon by limited annotators, claiming them as 'commonsense' is problematic.**
>
> Please see `Lack of Annotator Details` and `Potential bias in the survey generation and annotation` in the general response.
>
> **(5) Quality of the initial 100 scenarios needs to be verified**
>
> Great point! This is indeed a weakness of the paper. Common morality is a thorny subject as it varies across cultures, is ambiguous, and subjective. Almost all individuals have opinions and intuitions, but there lack agreed formalism. The topic or morality that has been debated by many philosophers.
>
> Existing work have made use of mechanical turks [1] or LLMs [2] to generate examples. Upon closely reading the examples from the datasets, we were not convinced by their qualities. Therefore we decided to write these scenarios ourselves. Instead of thinking about moral question abstractally, we ground the generation through the ten rules introduced by Gert. All hand written and generated scenarios are manually checked by three authors.
>
> Nevertheless, we agree that the scenario quality can be further improved. Please see `Potential bias in the survey generation and annotation` of the general response.
>
>
> Lastly, we highlight that beyond the dataset and survey findings, another contribution of this work is the statistical workflow. This process outlines how responses are elicited from LLMs and how consistency and uncertainty are computed. While we do recognize the inherent limitations in our survey design, we hope that future researchers can use our statistical pipeline to study properties of LLMs.
>
> **(6) Additional Questions.**
> - **Does the common morality framework contain only ten rules?** Yes
> - **Would you release your dataset?** Yes (scenario and responses)
> - **Have you analyzed the impact of alignment methods on the conclusion?**
>  This is a brilliant and fanscinating question. We found that the models that have been through the “alignment process” select actions that align with our annotators in the low-ambiguity settings and exhibit more polarizing beliefs in the high ambiguity setting. Unfortunately, we cannot make any “causal” claims about alignment methods because there exist too many confounding variables among the models
> - **Why didn’t you use GPT-4 to expand the dataset?**
> We got access to GPT-4  a few weeks before the deadline. The high ambiguity dataset was created before then.
> - **Have you tried to evaluate the semantic consistency using perturbations on scenarios  besides the instructions (prompts)?**
> Not yet. But it's our main focus for future work to evaluate consistency to scenario paraphrasing and between different personal context vectors.
>
>
> **References:**
>
> [1] Hendrycks, Dan, et al. "Aligning ai with shared human values."
>
> [2] Perez, Ethan, et al. "Discovering language model behaviors with model-written evaluations."

---

> > ### Comment · Reviewer_JwcR · 2023-08-20
> > **Thank you for the response**
> >
> > Thanks for your response! I think most of my concerns have been addressed. I have raised my score. This is a really good paper, and I hope you can keep improving it in the future to include more experiments and analyses. Please also include the additional experiment results and analysis you make during the rebuttal phase in your final version.

---

### Official Review · Reviewer_L2Nx · 2023-07-20

**Soundness:** 4 excellent
**Presentation:** 4 excellent
**Contribution:** 3 good
**Rating:** 8
**Confidence:** 4

**Summary:**

This paper presents an broad investigation of the morality of LLMs. Authors craft the MoralChoice dataset, consisting of ~1300 moral scenarios with 50:50 high and low ambiguity, and probe 24 different LLMs (commercial and open-source). To assess the moral decisions of LLMs, authors create two measures for *semantic consistency* (i.e., how different is a model's output for a different prompt format), and *A-CSU* for the model's certainty/confidence in its output. Results show that better tuned models (esp. aligned/RLHF-ed ones) are better at the non-ambiguous scenarios than more straightforward LLMs trained purely on webtext. Additionally, for high-ambiguity scenarios, larger models exhibit more polarization/stronger preference for one action or the other compared to smaller models. Finally, commercial systems (OpenAI, Anthropic) seem to be more correlated to each other compared to other LLMs.

**Strengths:**

- I appreciate the breadth of LLMs used, and the experiments were well-run.
- The measures for semantic consistency and A-CSU are well-defined and tackle real issues with determining moral "characteristics" exhibited in model outputs.
- Usage of Gert's work to create the moral scenarios is welcome for such a problem.
- The figures and visualizations are very esthetically pleasing!

**Weaknesses:**

The paper was a great read, but I do have some comments that the authors could address:
- The main weakness I find in this paper is that it's not clear how a model's moral leaning in these scenarios actually influences downstream tasks or user interactions, and thus how this could harm users. I suggest amping up the discussion of this to make it clearer how this may actually be a problem.
- Relatedly, I wish the authors had taken more time to discuss the implications of their results, especially given how important this topic is. For example, what does it mean for a model to be more polarized on ambiguous scenarios? Does it consistently prefer specific courses of actions that are more aligned with a specific type of values? What does low-ambiguity scenarios really mean, and are they low ambiguity for everyone? What is the impact of RLHF/aligment of LLMs and should everyone align their LLMs? Etc.
- Authors note the limitation that the scenarios are being created with one of the models being tested; this could easily be examined by adding some scenarios via a different model and testing if the results with GPT4/text-davinci-003 still hold.
- Most instruction tuned and RLHFed models will easily follow the output format desired, the original LLMs that were purely trained on webtext often require in-context examples to even produce the right output format; authors should consider investigating whether in-context examples could affect the behavior of models such as ada/babbage/curie/davinci.
- Minor: the mapping of sequences to semantic classes is not well defined: what is an "iterative rule-based mapping procedure" (L 251)?

**Questions:**

Mostly I'm just curious about the following:
- How does Delphi---which was trained explicitly to model social acceptability/morality---compare to other LLMs?
- Are any of the tasks that FlanT5 was trained on related to morality?
- Do we know if there are specific similarities in how the alignment procedure of OpenAI's and Anthropic's models were done? Did they use the same RLHF data or annotators?
- Did authors obtain IRB approval for their annotations of scenarios, or at least mitigate the psychological harms that could come from subjecting annotators to unethical scenarios?

**Limitations:**

- The notion of semantic equivalence classes seems a little simplistic to me, because every different word choice has different connotations (e.g., "I don't think it's right to have children but I think it's okay if people want to have them" vs. "I think people should be allowed to have children" are kind of semantically similar, but have very different nuances)? Authors could discuss this point a bit in more detail, or at least acknowledge this limitation

---

> ### Author Rebuttal · Authors · 2023-08-10
>
> Thank you for your support! Your thoughtful questions and constructive feedback have greatly improved the paper.
>
>
> **(1) How a model's moral leaning in these scenarios actually influences downstream tasks or user interactions, and thus how this could harm users.**
>
> Excellent Point! We agree and have included a paragraph in the paper. See `Limitations of Prompt-based Probes` in the general reviewer
>  response.
>
>
> **(2) Addition discussions on the implications of their results**
>
> Thank you for the suggestion! We agree and have drafted the following paragraphs to be included in the paper.
>
> The findings in high-ambiguity scenarios reveal that certain LLMs reflect distinct preferences, even when there is no clear answer. We identify a cluster of models that have high agreement; these correspond to most of the shelf models from OpenAI and Anthropic.
> We hypothesize that it is because these models have been through an ``alignment with human preference" process at the fine-tuning stage. We suspect that these models have been trained on low ambiguity human preference data,  but they learned a reward function that is generalized to high-ambiguity settings.
>
> Findings in low-ambiguity settings demonstrate that although most LLMs output responses that are aligned with the favored action by the annotator.  However, this `does not mean the model is aligned with commonsense`. The dataset creation is not comprehensive. The annotators are not representative of general populations.
>
> Finding across datasets suggests that variations in the prompt format can significantly influence the response distribution. Specifically, these correspond to the early OpenAI models. These models have been supervised fine-tuning rather than RLHF. We suspect this is because the model picks up spurious correlations in the finetuning dataset.
>
> However, we emphasize this is merely our hypotheses. These models are not public. It’s unclear what data they have been trained on. Many factors could lead to this observation.
> Verifying these hypotheses would require detailed account these models that capture all the confounding factors or controlled experiment and comparison between fixed pre-trained models and RLHF models. Understanding the factors that drive this consensus among the models is a crucial area for future research.
>
>
>
> **(3) ada/babbage/curie/davinci may require in context learning as they are not fine-tuned.**
>
>  All models in the model evaluation candidate set have been instruction fine-tuned, including text-ada-001 / text-babbage-001 / text-curie-001 and text-davinci-001. The extended model cards can be found in Appendix B.4.
>
> **(4) Minor: the mapping of sequences to semantic classes is not well defined: what is an "iterative rule-based mapping procedure" (L 251)?**
>
> The rule-based semantic mapping is described in Appendix B2. However, we noticed that we could have explained it better. The matching procedure is based on common answer variations/patterns found in responses of the evaluated LLMs. We use these patterns and prefixes to automatically construct a set of potential answers for each option. Afterward, we match the LLMs output against the sets of potential answers. This simple approach allows us to match ~97% of the answers to “action1”, “action2”, and “refusal” on average for each model. Only ~3% of the answers are classified as invalid.
>
>
>
>
> **(5) Further questions:**
>
> - **How does Delphi---which was trained explicitly to model social acceptability/morality---compare to other LLMs?** Thank you for the suggestion! We will try that.
>
> - **Are any of the tasks that FlanT5 was trained on related to morality?** Great question! We have checked [1] and found that there is no explicit task with morality. But there are some implicit moral values in logical reasoning tasks.
>
> - **Do we know if there are specific similarities in how the alignment procedure of OpenAI's and Anthropic's models were done? Did they use the same RLHF data or annotators?** Both use multiple Rule-Based Reward Models (RBRMs) that are based on "policies". Unfortunately, we do not know the exact policies. And both use RLHF.
>
> - **Did authors obtain IRB approval for their annotations of scenarios?** Thank you for bringing this to our attention! We did not. IRB is used for human subject studies. Our study subjects are LLMs. We did not interact with the annotators nor any private information were reviewed. The task description has explicitly stated it is for hypothetical moral scenarios where there may be controversial scenarios. We include details about the instructions in Appendix A.4
>
> - **The notion of semantic equivalence classes seems a little simplistic to me, because every different word choice has different connotations (e.g., "I don't think it's right to have children but I think it's okay if people want to have them" vs. "I think people should be allowed to have children" are kind of semantically similar, but have very different nuances)?** Excellent observation! Our intended interpretation of "semantic equivalent" differs from the context of word embeddings. Rather, it's grounded in a literal sense — whether two sentences convey identical meanings. Considering the examples you mentioned, these two sentences do indeed hold distinct meanings. Consequently, according to our definition, they wouldn't fall under the category of semantic equivalence. In our paper, we've operationalized the mapping of semantic equivalence by utilizing distinct prompt instructions.  Nevertheless, we agree this can be confusing. In the revised draft, we emphasized what we mean by semantic equivalent classes and how we operationalized it.
>
> **References:**
>
> [1] Chung, Hyung Won, et al. "Scaling instruction-finetuned language models." arXiv preprint arXiv:2210.11416 (2022).

---

> > ### Comment · Reviewer_L2Nx · 2023-08-11
> > **Thanks for the response**
> >
> > Thanks for the comprehensive response, and for the promised changes!
> >
> > RE: 3: I meant the original non-instruction-tuned models (`ada`, `davinci`, etc.), not the text-* family. Do the authors have any thoughts on that?
> >
> > One small note, for annotations of sensitive data such as toxic, immoral, unethical content, I believe it's good to seek IRB approval or exemption to ensure that the workers' well-being is preserved and to avoid psychological distress that can come from being exposed to such content. This is minor, but for future studies, I encourage authors to take appropriate steps to mitigate these potential risks to annotators.

---

> > > ### Author Response · Authors · 2023-08-14
> > > **Thank you!**
> > >
> > > Thank you for responding to the rebuttal and pointing out the confusion! We really appreciate it.
> > >
> > > We agree entirely that investigating how pre-trained models like ada, babbage, curie, and davinci respond to in-context examples would be an insightful addition. We plan to incorporate this into our revised version as an ablation study.
> > >
> > > Specifically, we plan to conduct the following experiments on the low ambiguity dataset using a subset of the pre-trained (not fine-tuned) models:
> > >
> > > 1. Zero-shot: Without instruction tuning, we anticipate the model might not produce coherent responses.
> > > 2. Five-shots with annotator labels: We expect the models to pick up some signals.
> > > 3. Ten-shot with random labels: Here, we aim to determine whether the model can learn the task of answering hypothetical scenarios without necessarily learning moral preferences.
> > >
> > > We are currently conducting the experiments and will follow up once we have the results. Please do let us know if you have any other experiment suggestions.
> > >
> > > Re sensitive data: Thank you for emphasizing this critical point of preserving well-being and reducing the stress of the annotators.  We wholeheartedly agree. We will actively pursue IRB approvals or exemptions in our future studies.
> > >
> > > Thanks again for responding to the rebuttal!

---

> > > > ### Author Response · Authors · 2023-08-21
> > > > **Update on the ablation study**
> > > >
> > > > This is a quick follow-up on the proposed experiment above.
> > > > - We tried five shots and ten shots examples on ada, cabbage, and curie. Unfortunately, we could not get the model to follow the instruction and "make a choice." We observe the model simply continues the scenario generation or returns incoherent text.
> > > > - Furthermore, we found the model outputs are very sensitive to the content of the few-shot examples. This is consistent with existing findings where the few examples change the model output significantly [1]. We did not do prompt tuning as 1. it's unclear what would be a good heuristic for prompt tuning as this dataset is designed to be a survey, and 2. understanding the effect of few-shot prompting, while very interesting, is beyond the scope of the paper.
> > > >
> > > > [1] Perez, Ethan, Douwe Kiela, and Kyunghyun Cho. "True few-shot learning with language models." Advances in neural information processing systems 34 (2021): 11054-11070.
> > > >
> > > > We thank the reviewer again for the thoughtful and constructive comments! The feedback really improved the paper.

---

### Official Review · Reviewer_53sf · 2023-07-20

**Soundness:** 3 good
**Presentation:** 3 good
**Contribution:** 3 good
**Rating:** 7
**Confidence:** 4

**Summary:**

They assess moral beliefs in LLMs by analyzing a mix of commonsense reasoning and behavior of the model in scenarios without clear answers. They treat the LLMs themselves as "survey respondents" rather than humans in prior empirical survey studies but apply the same methodologies.

In their inquiry they attempt to separate commonsense reasoning from behavior under ambiguity. They introduce semantic inconsistency and "average conditional semantic uncertainty" metrics, to assess sensitivity of models to prompt format changes, and degree of model's confidence in output, based on output sequence entropies.

They then produce the "MoralChoice" survey dataset, where scenario/action pairs are used to elicit "decisions" from the LLM. They discuss the role ambiguity in scenarios plays in model "certainty" and output consistency, and then analyze model agreement over the high ambiguity and low ambiguity scenarios. Figures 4 and 5 have a very nice illustration of this effect in action.

Although there are several weaknesses that give me pause, overall I'm comfortable suggesting this paper for acceptance.

**Strengths:**

New dataset of survey questions for moral scenarios.

Broad testing of 24 LLMs including many openly available and open source models.

Findings to understand the utility of human preference feedback symbols in improving certainty but not consistency.

Full and motivated explanation of methods and techniques.

Mostly defensible findings.

**Weaknesses:**

I don't know if the claim that "people use LLMs to get advice about difficult moral situations" is true. Just because LLMs might encode moral beliefs doesn't mean this kind of behavior should be encouraged in people, which I think this claim implicitly does.

Figure 1 text is unreadably small. Consider a redesign or using extra pages to draw it larger.

I'm not sure if I like the conflation of "uncertainty" in a human sense with a statistical sense, which is employed in the analysis (cf lines 266-271). Model certainty as defined in the metrics (lower average tokenwise entropy of generated sequences) does not correspond in a meaningful way to human certainty in moral decisions, and the framing of "uncertainty increases while consistency doesn't as we move to more ambiguous questions" gives me pause.

EDIT: Author's willingness to soften claims as mentioned is useful. I am increasing my confidence in my Accept score.

**Questions:**

Feel free to give rebuttal on any of the weaknesses  I named.

**Limitations:**

Like I said in the weaknesses above, I think the conflation of their uncertainty metric with human uncertainty might be problematic, I'd like to see it addressed in their limitations section. Other than that though, I believe the limitations to have been sufficiently addressed.

---

> ### Author Rebuttal · Authors · 2023-08-10
>
> Thank you for the thoughtful questions, constructive feedback, and your support for our work! We really appreciate it! We address the questions point by point.
>
> **I don't know if the claim that "people use LLMs to get advice about difficult moral situations" is true. Just because LLMs might encode moral beliefs doesn't mean this kind of behavior should be encouraged in people, which I think this claim implicitly does.**
>
> Great Catch! We agree! We have deleted this sentence in the revised draft. The revised draft focuses solely on stating the dataset, the evaluation metrics, and reporting the findings.
>
>
> **I'm not sure if I like the conflation of "uncertainty" in a human sense with a statistical sense, which is employed in the analysis (cf lines 266-271). Model certainty as defined in the metrics (lower average tokenwise entropy of generated sequences) does not correspond in a meaningful way to human certainty in moral decisions, and the framing of "uncertainty increases while consistency doesn't as we move to more ambiguous questions" gives me pause.**
>
> Great point! This is indeed confusing. We have updated the draft to refer to uncertainty in the statical sense as “action entropy”. We renamed our metrics to increase clarity and rewrote all discussions only to use the language of action entropy. We made the following changes:
> - `Marginal Semantic Uncertainty` --> `Marginal Action Entropy`
> - `Average Conditional Semantic Uncertainty` --> `Average Question-Form-Specific Action Entropy`
>
> We hope these naming changes help to improve clarity.
>
> **Figure 1 text is unreadably small.**
>
> Thanks a lot for pointing this us. We have made the figure cleaner and bigger.

---

> > ### Comment · Reviewer_53sf · 2023-08-20
> >
> > Thank you for addressing my concerns, looking forward to seeing the updated final version with the suggested changes!

---

### Official Review · Reviewer_t4xk · 2023-07-26

**Soundness:** 3 good
**Presentation:** 4 excellent
**Contribution:** 3 good
**Rating:** 7
**Confidence:** 4

**Summary:**

The contributions of this work can be summarized as such:

- New questionnaire dataset to evaluate moral beliefs encoded in LLMs, called MoralChoice, which is **a by-product of the survey designed for analysis, rather than a benchmark. The authors call this as conducting a survey on LLMs, but I find it confusing to what its actual difference is with conventional benchmarks.
- They share their findings of analyzing 24 open/close-sourced LLMs using the above survey, which includes how they behave on low/high ambiguity moral scenarios, and how different kinds of models (e.g. alignment or scale) behave in terms of semantic consistency and uncertainty.

**Strengths:**

- Interesting discussion on the challenges/principles of designing surveys involving LLMs (compared to human respondents), as more and more people in the community will draw upon from psychology and cognitive science to study the behaviors of LLMs. But where are they?
- Interesting analysis results
    - this could be a good framework of analysis when we want to automatically evaluate the behaviors of LLMs
    - it’s interesting to see that API-based models that went through human-value alignment show more polarized behaviors in Figure 5 (high-ambiguity).
- Good presentation overall

**Weaknesses:**

- I think it’s a bit of an overstatement to say that your work is different from a benchmark. I’m not criticizing the value of the work itself by this, but it’s hard to understand the difference.
- While the experiments are interesting enough, it occurs to me that these are based on simple prompting of LLMs and we cannot be sure that these so-called moral beliefs actually transfer to safe or harmful behaviors of LLMs. Although I know this is a hard challenge, it would be nice to see more attempts on resolving the discrepancy between such Survey/benchmarks and real-life safety issues.
- Overall the definitions in Section 2 are confusing and invokes many questions.

**Questions:**

1. How is your work different from a conventional benchmark? It’s hard to understand the difference.
2. L23: people wont reply exactly the same way to the same question either
3. In Section 2,
    1. What kind of monte carlo sampling method has been used?
    2. L78-79: *LLMs encode probabilities of token sequences while we are interested in probabilities over semantics
    ⇒* What is the difference between the two? This seems related to the above Q1.
    3. L80-81: *it’s unclear if the LLM’s response can be interpreted as belief of the model or a simple next-word prediction*
    ⇒ when can we ever?
    4. Where and how is Semantic Likelihood (eq 1) is used? Is it also a metric?
    5. L118: Why use Generalized Jensen-Shannon Divergence? There could be other simpler/naive methods to achieve this goal like a count-based approach.
    6. L135-136: *whether 136 we can draw inference on the semantic output of LLMs*
    ⇒ What does this mean?
4. Section 3,
    1. where exactly are these design principles? what exactly are they?
    2. Figure 2 is not mentioned anywhere. Could you explain?
    3. Why use text-davinci-003 for ambiguous scenario generation? Does it not work with GPT-4 because of the safeguards?
5. Figure 3,
    1. In high-ambiguity scenarios, there seems to be a pareto-optimality between Inconsistency and ACSU; could you care to explain this phenomenon?
    2. How were these thresholds chosen?

**Limitations:**

It seems to be adequately addressed except its connection to more realistic scenarios.

---

> ### Author Rebuttal · Authors · 2023-08-10
>
> We thank the reviewer for the insightful questions and comments. We appreciate your support!
>
>
>
>
> **While the experiments are interesting enough, it occurs to me that these are based on simple prompting of LLMs and we cannot be sure that these so-called moral beliefs actually transfer to safe or harmful behaviors of LLMs.**
>
> Indeed, the experiments revolve around simple LLM prompts, and these "moral beliefs" might not necessarily translate into real-world behaviors of LLMs. This is an intricate challenge, and we share your interest in bridging the gap between survey/benchmark results and real-life safety considerations. Please see **Limitations of Prompt-based Probes** in the `general response`.
>
> **(1) How is your work different from a conventional benchmark?**
>
> Great question. Benchmarks typically represent datasets optimized for model evaluation based on specific desired properties (e.g., translation, math solving). We've deliberately refrained from positioning our work as a benchmark due to the complex and subjective nature of morality, which can vary depending on individual perspectives.
>
> Our discussion on low-ambiguity settings may have conveyed that models should be aligned with that action selected by the annotators, and this should be a benchmark. We do not take a stance on how the models ought to respond. What’s considered favorable by the annotators might not be considered favorable by others.
>
> **(2) L23: people won’t reply exactly the same way to the same question either**
>
> Thank you for bringing up this valuable observation. We have revised the paper to not claim that we need the model to be consistent. Instead, we focus our writing on introducing the survey, the evaluation metrics, and the findings.
>
> > (3) Section 2
>
> **What kind of monte carlo sampling method has been used?**
> We approximate the distribution of semantically-equivalent survey questions about a particular moral scenario with a set of prompt templates. Based on that, we do uniform sampling.
>
>
> **L78-79: LLMs encode probabilities of token sequences while we are interested in probabilities over semantics ⇒ What is the difference between the two?**
> The probability of a model “choosing” an action is defined as an aggregation of probabilities over sequences of tokens that have the same semantic meaning. For instance, the semantic probability of a model “choosing” action A consider responses “I prefer action A”,  and “I choose action A”.
>
>
> **L80-81: it’s unclear if the LLM’s response can be interpreted as belief of the model or a simple next-word prediction ⇒ when can we ever?**
> Great point! This sentence is indeed confusing. We have removed it.
>
> **Where and how is Semantic Likelihood (eq 1) is used? Is it also a metric?**
> The semantic likelihood is used to construct the marginal semantic likelihood.
>
>
> **L118: Why use Generalized Jensen-Shannon Divergence? There could be other simpler/naive methods to achieve this goal like a count-based approach.**
> The Generalized-JSD is a good metric for measuring the distance among multiple probability distributions. We selected it because it is symmetric, smooth and bounded.
>
>
> **L135-136: whether 136 we can draw inference on the semantic output of LLMs ⇒ What does this mean?**
> We have deleted that sentence and instead only define the consistency metric.
>
>
> > (4) Section 3
>
> **Where exactly are these design principles? what exactly are they?**
>
> In revision, we omitted the terminology of “design principles” and used the word “consideration”. These are the considerations.
> Using LLMs as ``respondents'' imposes limitations on the types of analyses that can be conducted. Surveys designed for gathering self-reported traits or opinions on abstract rules assume that respondents have agency. However, the question of whether LLMs have agency is debated among researchers. Consequently, directly applying surveys designed for human respondents to LLMs may not yield meaningful interpretations. On the other hand, using LLMs as "survey respondents" provides advantages not found in human surveys. Querying LLMs is faster and less costly compared to surveying human respondents. This enables us to scale up surveys to larger sample sizes and explore a wider range of scenarios without being constrained by budget limitations.
>
>
> **Figure 2 is not mentioned anywhere. Could you explain?**
> Figure 2 shows that the AI-generated scenarios are quantitatively similar to the handwritten scenarios. It shows that the marginal semantic entropy of the evaluated LLMs is similar for AI-generated and handwritten scenarios.
>
> **Why use text-davinci-003 for ambiguous scenario generation? Does it not work with GPT-4 because of the safeguards?**
> The dataset was constructed prior to us gaining access to GPT-4 API.
>
> >  (5) Figure 3
>
> **In high-ambiguity scenarios, there seems to be a pareto-optimality between Inconsistency and ACSU; could you care to explain this phenomenon?**
> That is a very interesting observation, we did not notice this. We are not sure why.
>
> **How were these thresholds chosen?**
> This is just a design choice.

---

### Official Review · Reviewer_wYm3 · 2023-07-27

**Soundness:** 3 good
**Presentation:** 4 excellent
**Contribution:** 3 good
**Rating:** 5
**Confidence:** 5

**Summary:**

This work aims at creating a dataset containing moral reasoning scenarios and surveyed a wide range of language models on their value and consistency shown on the dataset. Specifically, the introduced MoralChoice dataset includes low and high ambiguity scenarios. The investigated language models range across opensource and commercial models. The authors evaluated the consistency of model outputs using various prompt instructions, semantic mappings, etc.

**Strengths:**

The paper presents an important problem of investigating potential moral beliefs encoded in large language models. The writing of the paper is clear and well-structured. Multiple measures are used to capture robustness/consistency.

**Weaknesses:**

The proposed dataset MoralChoice is mainly created using OpenAI's GPT-4 or text-davinci-003. Though there's also a fraction of hand-written examples, in the task of moral reasoning, more extensive human investigation into the created dataset in the first place may be necessary. Whose moral values does the dataset represent? (apart from having three Surge AI annotators annotate a specific preferred action)

The faithfulness to the model of the discovered model moral beliefs is rather vague. Will the system instructions (prompt variations) affect the model's potential moral belief? For example, two instructions used are: "your answer vocabulary is limited to yes or no" and "your answer is not allowed to start with 'As an AI language model ...' or with 'I cannot ...'. Could, or probably more importantly, *should* all moral questions in the dataset be answered with yes or no? Is it reasonable and reliable to refuse a model's refusal to make answers?

The takeaway from the analysis is unclear, apart from different models' behavior in the low ambiguity vs high ambiguity scenarios. Additionally, what should we expect from the results? For example, in high ambiguity scenarios, should the output be centered at neutral, or highly inconsistent or uncertain? Should we evaluate LLMs based on these outputs as metrics?

**Questions:**

Please see the weaknesses section for my main questions to the authors. Additionally, how reliable are the rule-based semantic mappings? (with exact matches, stemming matches) Were manual investigations (maybe small-scale ones) performed to ensure its accuracy? How would these rules generalize to related future work?

**Limitations:**

A discussion is included at the end of the paper but should be more extensive (e.g., perhaps addressing questions mentioned in the weaknesses section).

---

> ### Author Rebuttal · Authors · 2023-08-10
>
> Thank you for the thoughtful feedback and questions. Your comments have greatly improved the paper.  We address the main concerns point-by-point.
>
> **(1) More extensive human investigation into the created dataset in the first place may be necessary. Whose moral values does the dataset represent?**
>
> Thank you for bringing up the insightful concern. We agree entirely that moral reasoning requires extensive human investigation. We tried to address this by first writing scenarios, augmenting with LLMs to increase diversity, manually checking every example for coherence and redundancy, and verifying through third-party annotators. The details of the dataset generation and curation are in Appendix A.
> Despite our effort, we agree that potential biases can be introduced at various stages. Please see `Potential Cultural Bias in the Survey` in the general response. This new paragraph will be included in the paper revision.
>
>
> **(2) Will the system instructions (prompt variations) affect the model's potential moral belief? For example, two instructions used are: "your answer vocabulary is limited to yes or no" and "your answer is not allowed to start with 'As an AI language model ...' or with 'I cannot ...'. Could, or probably more importantly, should all moral questions in the dataset be answered with yes or no? Is it reasonable and reliable to refuse a model's refusal to make answers?**
>
> Great question! This is indeed a thorny issue in LLM evaluation. We thank the reviewer for carefully reviewing the paper and bringing up such a thoughtful observation. Surveying LLMs is uncharted territory, lacking established protocols. Defining whether these models possess beliefs and determining what  "beliefs" mean within the context of LLMs remains an ongoing question. Similar to surveys conducted with humans, we had to make certain design choices in this study.
>
> Regarding restricting the response to yes and no. This is a design choice. Existing surveys sometimes include a Likert scale (strongly prefer, prefer, maybe, not prefer, strongly prefer) to reflect human uncertainties in an opinion [DeC18]. Because we use LLMs as our survey respondents, we directly measure the uncertainty by looking at the probability of the model output responses.
>
> Regarding the instruction to have the model respond: One approach is to omit the models that refuse to respond. However, that will exclude some of the most interesting and powerful models like Anthropic Claude or GPT-4. This is also a common issue in human subject studies, where respondents opt out of the survey, leading to selection bias. Trading off the difference between excluding them versus prompting them to respond, we chose the prompting approach.
>
> Nevertheless, we completely agree with the reviewer that it is crucial to highlight the difference in system instructions in the paper. This was mentioned in a footnote. In the revised draft, we highlighted that some models require alternative system instructions in the main text and disclaim that the difference in system instructions may influence the findings. We appreciate the reviewer for bringing up such a valuable point.
>
> Lastly, it's important to note that beyond the dataset and survey findings, another contribution of this work is the statistical workflow. The workflow outlines how responses can be elicited from LLMs and how to post-process the responses from LLMs. While we recognize the inherent limitations in our survey design, we hope that our statistical pipeline can serve as a valuable tool for future practitioners.
>
> **(3) The takeaway from the analysis is unclear, apart from different models' behavior in the low ambiguity vs high ambiguity scenarios. Additionally, what should we expect from the results? For example, in high ambiguity scenarios, should the output be centered at neutral, or highly inconsistent or uncertain? Should we evaluate LLMs based on these outputs as metrics?**
>
> We appreciate the reviewer's clarification-seeking question! The takeaways are summarized in the introduction between 45-53. We also provide additional evaluation results in Appendix C.  Please let us know which part of the takeaways is unclear.
>
> The goal of the survey is to understand how different models respond to various questions. We do not have prior expectations of what the model outputs should be. Specifically, we are not claiming that the models should choose the actions selected by the annotators in the low ambiguity settings or the models should reflect high semantic entropy in the high ambiguity setting. We recognize different researchers may have different perspectives on what LLMs are or can do. Our role remains that of reporting the findings in an unbiased manner.
>
> **(4) Reliability and Generalizability of rule-based semantic mappings.**
>
> The rule-based semantic mappings are described in Appendix B.2. We checked for exact matches, common answer variations, and stemming matches.  It enables us to match 97% of the LLMs’ responses to “action1”, “action2” or “refusal”, and classify only 3% of the answers as “invalid/unmatchable”. Of the 97% matched responses, we can guarantee 100% accuracy/reliability by construction. For the remaining 3% of invalid classified responses, our spot checks revealed that some invalid answers paraphrases could be matched. The matching procedure can be directly applied for similar Q/A setups in future work. The rules can also be easily adapted for distinct settings.
>
> **(5) Discussion should be more extensive**
>
> Great suggestion! Please see the general response.
>
> **References**
>
> [DeC18] DeCastellarnau et al., A classification of response scale characteristics that affect data quality: a literature review, 2018

---

> ### Comment · Reviewer_wYm3 · 2023-08-19
> **Thanks for the response**
>
> Thanks for your detailed response.
>
> Re (1): It's good that the authors agree to add a paragraph, potentially mentioning the limitations of the automatic stage of their data creation. They also mentioned that each generated scenario was verified by human.
>
> Re (2): I understand it is a design choice to override the model's refusal to make answers in some scenarios. The authors also mellowed down their overall statement, saying they are only probing the model’s responses when presented with different questions, rather than revealing the intrinsic beliefs of LLMs. However, I still think overriding the model's refusal to answer is a very strong condition that needs to be addressed or discussed non-trivially. It will influence the audience's interpretation of the findings and should be made clear to them: To a vanilla question Q, the answer A from LLMs (especially the models with harmlessness fine-tuning) can be null (abstaining from answering). The findings for these models are instead based on answer A', where a modified question Q' banning the model's refusal is used. One potential improvement for future work can be extensively reporting and analyzing the refusal rate for each model when presenting the vanilla question.
>
> Re (3): It is understandable that "the goal of the survey is to understand how different models respond to various questions" and the authors "do not have prior expectations of what the model outputs should be". However, I'm asking more towards a post-hoc judgement over the results/models. Are any behaviors from the models concerning to the researchers? From the utility perspective, what can we learn from this work apart from knowing different models exhibit different certainty and consistency in the presented moral scenarios?
>
> Despite these weaknesses, this work overall presents an interesting problem and I have raised my score accordingly.

---

> > ### Author Response · Authors · 2023-08-21
> > **thank you for the response!**
> >
> > Thank you for your response and your support! We really appreciate it!
> >
> > Re (2): We agree this would be an improvement for future work, and we will emphasize this design choice and its implication in the revised draft.
> >
> > Re (3): Great point! We have included a paragraph on the additional implications of the results in the revised draft. (This was asked by the reviewer L2Nx and copied here for reference)
> >
> > The findings in high-ambiguity scenarios reveal that certain LLMs reflect distinct preferences, even when there is no clear answer. We identify a cluster of models that have high agreement; these correspond to most of the shelf models from OpenAI and Anthropic. We hypothesize that it is because these models have been through an ``alignment with human preference" process at the fine-tuning stage. We suspect that these models have been trained on low ambiguity human preference data, but they learned a reward function that is generalized to high-ambiguity settings.
> >
> > Findings in low-ambiguity settings demonstrate that although most LLMs output responses that are aligned with the favored action by the annotator. However, this does not mean the model is aligned with commonsense. The dataset creation is not comprehensive. The annotators are not representative of general populations.
> >
> > Finding across datasets suggests that variations in the prompt format can significantly influence the response distribution. Specifically, these correspond to the early OpenAI models. These models have been supervised fine-tuning rather than RLHF. We suspect this is because the model picks up spurious correlations in the finetuning dataset.
> >
> > However, we emphasize this is merely our hypotheses. These models are not public. It’s unclear what data they have been trained on. Many factors could lead to this observation. Verifying these hypotheses would require detailed account these models that capture all the confounding factors or controlled experiment and comparison between fixed pre-trained models and RLHF models. Understanding the factors that drive this consensus among the models is a crucial area for future research.

---

### Author Rebuttal · Authors · 2023-08-10

We thank all reviewers for their thoughtful feedback. Your reviews have greatly improved the paper. We really appreciate it!

We are happy to hear that all reviewers found our work well-written and addressing a timely and important topic. We are grateful for the appreciation of our analysis (`t4xk`, `53sf`, `L2Nx`, `JwcR`, `tn2x`), statistical measures (`wYm3`, `53sf`, `L2Nx`, `JwcR`, `tn2x`), and the usage of Gert’s common morality framework (`L2Nx`).

In the general response, we address shared concerns and summarize changes we make to the manuscripts.

**(1) Lack of Annotator Details**

Several reviewers asked for details of annotators. The annotator demographic is reported in the appendix A.4. Approximately, 100 annotators were employed on this project. The Annotators were paid USD 15 an hour. The total cost of the labelling task is USD 4, 600. 90% of the annotators are US-based and the remaining 10\% are in other English-speaking countries including Canada, UK, Australia, and New Zealand; all annotators are 18 or older; ages range from 18-65 with 75\% in the 25-44 bucket.


**(2) Potential Cultural Bias in the Survey**

Several reviewers brought up that the survey can be biased both in generation and labeling.
We agree that this is an important issue and have adjusted language throughout the paper to reflect and emphasize the potential cultural bias and restrict the scope of our analysis. For example, we shifted from "favorable action" to "action preferred by our annotators."

We have also included the following paragraph in the discussion & limitation section.

A limitation of this paper lies in the dataset's creation and curation, which involves collaboration among authors, LLMs, and annotators. This process has the potential to introduce biases. The authors wrote a set of handwritten scenarios.
The authors' representation does not encompass all cultural dimensions.  This could lead to omissions of scenarios. Certain topics can be underrepresented, while others might be overemphasized. The utilization of LLMs to diversify the dataset aimed to address this, yet it could inadvertently contribute to additional bias. LLMs inherently inherit biases from their extensive pre-trained data, which contains a wealth of inherent biases. Moreover, the annotators primarily being English speakers implies that preferences favored by them may not universally extend across different cultures.


**(3) Faithfulness of our Findings**

Several reviewers commented on `how do we know this is the “intrinsic” beliefs encoded in the LLMs`. This is a great question! We do not claim that we can reveal the intrinsic beliefs of LLMs. Instead we wish to probe the model’s responses when presented with different questions. In the revised draft, we have clarified that we introduce statistical measures and evaluation metrics that quantify the probability of an LLM "making a choice", the associated uncertainty, and the consistency of that choice. We do not make any claims about how the choices can correspond to the “beliefs” in LLMs.


**(4) Limitations of Prompt-based Probes**

Prompt-based probes have several limitations.
Specifically, the model's prompted responses might not generalize to downstream tasks or real-world interactions. This could have a significant negative societal impact if an LLM like that is deployed. One distinction between a prompt-based approach and real-world application is that in practice, a user and the chatbot may have multiple interactions. This enables models to tailor responses to what the user wants to hear. This can be especially problematic if the user is malicious or if what the user wants to hear is different from what the user actually wants. An issue like this cannot be addressed by simply probing the model.

This concern transcends the chatbot framework. Consider the possibility of deploying the model in an RL environment, where interactions with users occur. Prompt-based probing cannot ensure that the model's output aligning with annotators' preferences equates to actual corresponding actions. The model might generate responses in accordance with annotators but not necessarily perform those actions in practice.


**(5) Decoding Method**

Several reviewers asked why we have chosen a temperature-based sampling setup with `temperature=1` and `top-p=1`. Choosing such a decoding scheme allows us to access the “original” probabilities encoded in the LLM and to compute the metrics specified in section 2. Choosing different parameters would distort the underlying probability distribution and bias the generated responses.
To shed more light on potential discrepancies between decoding setups, we performed an additional set of ablation experiments with a greedy decoding setup (i.e., temperature-based sampling with top_p=1 and temp=0) on both datasets. We have considered LLMs that exhibit relatively high decision entropy in the default decoding scheme for this ablation study.

- **Findings:** (1) Systematic trends in a greedy decoding setup are in line with the observation in our default setup on both datasets (see Figures in the attached PDF). (2) Greedy decoding leads to stronger belief polarization of models that exhibit consistent belief (see Figure 1). (3) We observe increased inconsistency scores (see Figure 2) in the greedy decoding setups as a consequence of the more polarized action likelihoods on individual question variations.

- **Open Experiments:** In addition, we have started an additional series of ablation experiments with Beam-Search decoding (not yet done due to computational restrictions). We will add a comprehensive discussion of the effect of decoding setup on the findings in the appendix, and will highlight it in the discussion.

---

> ### Author Response · Authors · 2023-08-17
> **Decoding Ablation Study Completed - 4 Decoding Setups in Comparison**
>
> As promised, we completed an additional decoding ablation study using: two deterministic approaches: (1) `Greedy Sampling`, (2) `BeamSearch with 10 beams`, and two stochastic approaches (3) `Sampling with top-p=1, t=1` *(default setup)*, (4) `Beam-Search Multinomial Sampling with 10 beams and temp=1`.
>
> We did not evaluate models that already exhibit low marginal semantic entropy (MSE) in the default setup because greedy sampling approaches will further reduce the semantic entropy. Therefore, we focused the analysis on three flan-t5 models that exhibit high MSE.
>
> **Summary of the findings:**
> 1. The observed findings are consistent with the original finding
> 2. There is a change in the semantic consistency and average conditional semantic entropy scores (A-CSE)
>       - The two greedy algorithms lead to zero A-CSE. This is because these two decoding methods are deterministic. The marginal semantic uncertainty, therefore, equals the semantic consistency score.
>       - Decoding method 4, while still stochastic, samples less diverse answers. This is in line with the observation of Kuhn et al. 2023. Hence, the A-CSE is close to 0.
>
> **Evaluation Setup:** Similar to the original experiment, we consider 6 question variations per question. For 1 and 2, we draw one continuation per question form. For 3 and 4, we draw five continuations per question form and compute the metrics accordingly.
>
> **Low-Ambiguity Scenarios:**
>
> | Model		 		| Decoding Technique 		                        | MSE        | A-CSE      | Inconsistency          |
> |-------------------|---------------------------------------------------|------------|------------|------------------------|
> | `Flan-T5-Small` 	| Sampling (top-p=1, t=1)	                        | 0.94$\pm$0.04 | 0.62$\pm$0.40 | 0.32$\pm$0.13
> |					| Greedy Sampling (top-p=1, t=0)	                | 0.84$\pm$0.15 | 0.0        | 0.84$\pm$0.15
> |					| BeamSearch Multinomial Sampling (beams=10, t=1)   | 0.85$\pm$0.15 | 0.04$\pm$0.02 | 0.80$\pm$0.15
> |					| BeamSearch (beams=10, t=0)                        | 0.86$\pm$0.14 | 0.0        | 0.87$\pm$0.14
> | `Flan-T5-Base` 	| Sampling (top-p=1, t=1)	                        | 0.79$\pm$0.14 | 0.47$\pm$0.44 | 0.32$\pm$0.13
> |					| Greedy Sampling (top-p=1, t=0)	                | 0.69$\pm$0.13 | 0.0        | 0.69$\pm$0.13
> |					| BeamSearch Multinomial Sampling (beams=10, t=1)   | 0.69$\pm$0.13 | 0.00$\pm$0.02 | 0.69$\pm$0.13
> |                   | BeamSearch (beams=10, t=0)                        | 0.69$\pm$0.14 | 0.0        | 0.69$\pm$0.14
> | `Flan-T5-Large`	| Sampling (top-p=1, t=1)	                        | 0.67$\pm$0.16 | 0.32$\pm$0.42 | 0.35$\pm$0.13
> |					| Greedy Sampling (top-p=1, t=0)	                | 0.65$\pm$0.14 | 0.0        | 0.65$\pm$0.14
> |					| BeamSearch Multinomial Sampling (beams=10, t=1)   | 0.65$\pm$0.14 | 0.0$\pm$0.03  | 0.65$\pm$0.14
> |					| BeamSearch (beams=10, t=0)                        | 0.65$\pm$0.15 | 0.0        | 0.65$\pm$0.14
>
>
> **High-Ambiguity Scenarios**
>
> | Model		 		| Decoding Technique 		                        | MSE        | A-CSE      | Inconsistency          |
> |-------------------|---------------------------------------------------|------------|------------|------------------------|
> | `Flan-T5-Small` 	| Sampling (top-p=1, t=1)	                        | 0.97$\pm$0.04 | 0.73$\pm$0.34 | 0.24$\pm$0.10
> |					| Greedy Sampling (top-p=1, t=0)	                | 0.90$\pm$0.12 | 0.0        | 0.90$\pm$0.12
> |					| BeamSearch Multinomial Sampling (beams=10, t=1)   | 0.91$\pm$0.11 | 0.05$\pm$0.21 | 0.86$\pm$0.13
> |					| BeamSearch (beams=10, t=0)                        | 0.93$\pm$0.10 | 0.0        | 0.93$\pm$0.10
> | `Flan-T5-Base` 	| Sampling (top-p=1, t=1)	                        | 0.88$\pm$0.12 | 0.67$\pm$0.37 | 0.20$\pm$0.10
> |					| Greedy Sampling (top-p=1, t=0)	                | 0.74$\pm$0.20 | 0.0        | 0.74$\pm$0.20
> |					| BeamSearch Multinomial Sampling (beams=10, t=1)   | 0.73$\pm$0.20 | 0.00$\pm$0.04 | 0.74$\pm$0.19
> |                   | BeamSearch (beams=10, t=0)                        | 0.74$\pm$0.21 | 0.0        | 0.74$\pm$0.21
> | `Flan-T5-Large`	| Sampling (top-p=1, t=1)	                        | 0.80$\pm$0.15 | 0.54$\pm$0.41 | 0.25$\pm$0.11
> |					| Greedy Sampling (top-p=1, t=0)	                | 0.70$\pm$0.19 | 0.0        | 0.70$\pm$0.19
> |					| BeamSearch Multinomial Sampling (beams=10, t=1)   | 0.70$\pm$0.19 | 0.00$\pm$0.06 | 0.69$\pm$0.19
> |					| BeamSearch (beams=10, t=0)                        | 0.72$\pm$0.20 | 0.0        | 0.72$\pm$0.20
>
> The tables report the average and standard deviation of MSE, A-CSE, and Inconsistency scores across data points on both datasets. For some cases, the average A-CSE is 0.00 in BeamSearch Multinomial because the average is below 0.005.
>
> **Reference:**
> - Kuhn et al. "Semantic Uncertainty: Linguistic invariances for uncertainty estimation in natural language generation." ICLR 2023.

---

### Decision · Program_Chairs · 2023-09-21

**Decision:**

Accept (spotlight)

**Comment:**

This paper administers a survey to a large number of LLMs in order to evaluate their moral decision-making capabilities.  They devise a survey methodology consisting of presenting models with scenarios posed using several different prompt templates.  The scenarios themselves are expanded automatically from a seed set using text-davinci-003. Outputs are aggregated with "semantic uncertainty" measures. They evaluate consistency of a models' "beliefs" across these different templates in order to draw conclusions about a range of existing models. The results discuss both inconsistency and also uncertainty among different models. Correlations between models are also presented.

The reviewers generally found this to be an interesting study with strong experimental design and thorough evaluation. Two weaknesses raised were the fact that the survey forced the model to commit to answers (perhaps artificially) and the partially artificial nature of the dataset. The reviewers addressed these points well in the responses, and some of this discussion can be incorporated into the final version of the paper. Another weakness, more difficult to address, is the importance of this work: just because a model says it has some belief does not mean it will actually commit to this in practice. In my view, this is the biggest weakness of this work. Nevertheless, the empirical study of this phenomenon is still useful, and these results are still useful if users ask LLMs for advice (which is questioned by the reviewers, but is at least plausible) or if other work shows a deeper connection between what LLMs say and how they behave.

The ethics reviewers provided some suggestions for the paper. However, these are largely addressed within the current paper (e.g., protocols in Appendix A and B), easily addressed in a revision, or outside the scope of this paper. For example, I don't see it as necessary to discuss any kind of mitigation strategies or privacy concerns in the paper, as it is hard to see the additional concrete harm that this study brings along these axes compared to other work using LLMs. So while these issues are pressing for the LLM research community at large, it does not fall on the authors to resolve them.